# Molecular Signatures in Swine Innate and Adaptive Immune Responses to African Swine Fever Virus Antigens p30/p54/CD2v Expressed Using a Highly Efficient Semliki Forest Virus Replicon System

**DOI:** 10.3390/ijms24119316

**Published:** 2023-05-26

**Authors:** Mei Huang, Hanghui Zheng, Weixiong Tan, Chengwei Xiang, Niran Fang, Wenting Xie, Lianghai Wen, Dingxiang Liu, Ruiai Chen

**Affiliations:** 1Zhaoqing Institute of Biotechnology Co., Ltd., Zhaoqing 526238, China; huangmeugz@163.com (M.H.); lukezheng@vectorbuilder.net (H.Z.); tanwx3586@163.com (W.T.); q18320727801@126.com (N.F.);; 2Zhaoqing Branch Center of Guangdong Laboratory for Lingnan Modern Agricultural Science and Technology, Zhaoqing 526000, China; xiangchengwei92@sina.cn; 3College of Veterinary Medicine, South China Agricultural University, Guangzhou 510642, China; 4Integrative Microbiology Research Centre, South China Agricultural University, Guangzhou 510642, China

**Keywords:** African swine fever virus (ASFV), innate immunity, adaptive immune response, differentially expressed genes (DEGs), weighted correlation network analysis (WGCNA)

## Abstract

African swine fever virus (ASFV) causes a devastating viral hemorrhagic disease in domestic pigs and Eurasian wild boars, posing a foremost threat to the swine industry and pig farming. The development of an effective vaccine is urgently needed, but has been hampered by the lack of an in-depth, mechanistic understanding of the host immune response to ASFV infection and the induction of protective immunity. In this study, we report that immunization of pigs with Semliki Forest Virus (SFV) replicon-based vaccine candidates expressing ASFV p30, p54, and CD2v, as well as their ubiquitin-fused derivatives, elicits T cell differentiation and expansion, promoting specific T cell and humoral immunity. Due to significant variations in the individual non-inbred pigs in response to the vaccination, a personalized analysis was conducted. Using integrated analysis of differentially expressed genes (DEGs), Venn, KEGG and WGCNA, Toll-like receptor, C-type lectin receptor, IL17 receptor, NOD-like receptor and nucleic acid sensor-mediated signaling pathways were demonstrated to be positively correlated to the antigen-stimulated antibody production and inversely correlated to the IFN-γ secreting cell counts in peripheral blood mononuclear cells (PBMCs). An up-regulation of CIQA, CIQB, CIQC, C4BPA, SOSC3, S100A8 and S100A9, and down-regulation of CTLA4, CXCL2, CXCL8, FOS, RGS1, EGR1 and SNAI1 are general in the innate immune response post-the second boost. This study reveals that pattern recognition receptors TLR4, DHX58/DDX58 and ZBP1, and chemokines CXCL2, CXCL8 and CXCL10 may play important roles in regulating this vaccination-stimulated adaptive immune response.

## 1. Introduction

African swine fever (ASF), caused by ASF virus (ASFV), is a devastating viral hemorrhagic disease in both domestic pigs and Eurasian wild boars. ASFV infects cells of the mononuclear phagocytic system, including differentiated macrophages and specific lineages of reticular system cells [1,2,3]. The highly virulent strains induce extensive damage in affected tissues with symptoms including hemorrhage, edema, ascites, and shock [4,5]. The development of an effective vaccine is urgently required for the prevention and control of this disease [6,7].

Different formats of ASFV vaccine candidates, including inactivated, live-attenuated (LAV), and subunit vaccines, have been attempted, but no effective and safe vaccine has yet been developed. Inactivated vaccines are immunogenic but usually not protective. LAV is protected against ASF effectively, but is generally viral strain-dependent and incomplete, being affected by multiple factors, such as dose and route of immunization, host immune status, and/or co-infection with another pathogen [6,7]. Vaccination of pigs with inefficient LAV might compromise the enforcement of other biosafety measures, leading to a fast viral spread [6,7]. Development of chronic ASF post-LAV vaccination was also reported in pigs vaccinated with a Portuguese ASFV attenuated by serial passaging in bone marrow cell cultures, a naturally attenuated ASFV isolates, ASFV/NH/P68, and a vaccine candidate, OUR T88/3, respectively [7,8,9]. Subunit vaccines including many ASFV structural proteins have also been tested in challenge studies, but their protection efficacies are inconsistent due to a variety of factors including different types of vaccine design, vaccination strategies, challenge strains, and animal genetic background [6,10,11,12,13]. Neutralizing antibodies against ASFV structural proteins may partially prevent animals from infection and delay disease development, but are unable to completely block viral replication and spread [14,15,16,17,18]. Essential in clear intracellular viruses, cellular immunity plays a key role against ASFV infection, with T cell response particularly crucial [19,20,21].

DNA vaccines have the advantage of inducing both cellular and humoral immunity without the risk of LAV, and can be constructed with many encoded safety features while retaining the specificity of a subunit vaccine [22]. Although the development of DNA vaccines had been hampered by poor product performance and low immune antigenicity, improvements can be made by gene optimization strategies, improved RNA structural design, more effective delivery approaches, and adding immune modulatory genes in the vaccine cocktail [22]. In fact, a few DNA vaccines have been successfully licensed including four veterinary vaccine products for horse, dog, pig and fish. Early DNA vaccines against ASFV were not successful in providing protective immunity, but partial protection of immunized pigs from lethal challenge with ASFV was achieved by adding a ubiquitin tag fused to the same antigen even in the absence of specific antibody response [21,23]. Ubiquitin may possibly improve the class I antigen presentation, as ubiquitination-mediated cytosolic proteasome degradation is a key factor in both conventional antigen processing and cross-presentation to generate MHC-I peptide complexes in antigen-presenting cells (APC) to prime naive CD8+ T lymphocytes. Macrophages and dendritic cells (DCs) are professional APCs and equipped with a vast array of pathogen recognition receptors (PRRs) to detect highly conserved microbial molecular signatures, and pathogen-associated molecular patterns (PAMPs). These cells are responsible for bridging innate and adaptive immunity.

ASFV has a large genome of approximately 190 kilobase pairs and encodes more than 150 proteins, including factors that modulate the host immune response and about half with no known or predictable functions [5,6]. Among ASFV structural proteins, p30, p54, and CD2v are well studied. Both p30 and p54 are involved in ASFV attachment and contribute to the antibody-mediated protective immune response [24]. CD2v, an outer-envelope protein of ASFV virion, is similar in structure and functions to the T-lymphocyte surface antigen CD2 on T and NK cells [25]. Deletion of CD2v coding gene *pEP402R* from the BA71 virulent strain attenuated the virus [12]. Anti-CDv2 antibody possesses neutralizing activity and partially protected immunized pigs from lethal infection [26].

With the rapid development of “omics” and systems biology during the past two decades, molecular and cellular signatures associated with the protection and immunogenicity of vaccines against different pathogens have been identified [27,28,29]. For example, the complement protein C1qB, the eukaryotic translation initiation factor 2 alpha kinase 4 EIF2AK4, and B cell growth factor TNFRS17 have been identified as key molecules for the induction of protective adaptive immune response by human yellow fever vaccine [27]. Early induction of kinase CaMKIV was inversely correlated with later antibody titers in humans vaccinated for seasonal influenza [28]. However, such studies are currently missing in ASFV research and vaccine development. An in-depth understanding of mechanisms controlling the host immune response to ASFV and identification of genes contributing to the induction of adaptive immunity by vaccine candidates would be of great help in the rational design and development of an effective ASFV vaccine.

Semliki Forest virus (SFV)-based replicon is derived from SFV, a member in the Alphavirus genus with a positive-stranded RNA genome of approximately 12 kb. The genome contains two main open reading frames (ORF), encoding a viral replicase and structural proteins, respectively. Replacement of the ORF coding for structural proteins by foreign genes creates an SFV-based vector system with a broad host range and high expression efficiency through self-driven replication. As this replication is self-limited due to the lack of viral structural proteins during the replicon package, it is an efficient and safe platform for foreign gene expression and vaccine development. In this study, we construct the SFV replicon-based DNA vaccine candidates expressing ASFV p30, p54, and CD2v proteins with or without fusion to ubiquitin and immunize small groups of non-inbred pigs in the form of DNA or infectious replicon particles (IRP). Comparative studies and personalized analysis on the induction of humoral and cellular immune responses and bioinformatics analysis of transcriptomic data identified genes and signaling pathways relevant to this vaccination-stimulated innate immunity and adaptive immune responses in pigs.

## 2. Results

### 2.1. Antigen Expression, Immunization Scheme, and Assessment

The N-terminal Flag-tagged p30, p54, and CD2v (or CD2vED) proteins with or without fusion to ubiquitin were expressed using SFV replicon-based plasmids, pSFV-flagp54, pSFV-flagCD2v, pSFV-flagp30, pSFV-flagUbiCD2vEDp30 and pSFV-flagUbiCD2vEDp54 (Appendix A). The expression efficiency of these proteins was checked by Western blot, demonstrating their efficient expression with expected molecular masses and reactivities with anti-ASFV sera (Figure 1a) as well as with an anti-Flag antibody. The ubiquitin-fused CD2vEDp30 and CD2vEDp54 were detected as multi-sized bands, indicating their modification by ubiquitination and ubiquitin-mediated protein cleavage, respectively (Figure 1a).

Co-transfection of BHK21 cells with pSFV-flagP54, pSFV-flagCD2v, pSFV-flagP30, pSFV-flagUbiCD2vEDp30, and pSFV-flagUbiCD2vEDp54, respectively, together with pCMV-SFV-helper 1 plasmid that expresses SFV structural proteins for a package of the recombinant SFV genomes, was then carried out to generate IRPs. The expression of target proteins in cells infected with IRPs was confirmed by immunofluorescent microscopy, showing high expression of p30, p54, and CD2v and moderate expression of UbiCD2vEDp30 and UbiCD2vEDp54, respectively (Figure 1b).

The replicon-based plasmids and IRPs were then used to immunize pigs. As described in the Material and Method section, group1 pigs (#1, #2, #5, and #6) were injected with plasmids or IRPs which express the target proteins without fusion to ubiquitin, while group 2 (#7, #8, #11 and #12) were injected with plasmids or IRPs which express the ubiquitin-fused proteins. All experimental pigs were immunized with two boosts after priming at a two-week interval, and whole blood samples were collected (Figure 1c). On day 28 (second boost) and day 29 (24 h post-second boost), an extra whole blood sample was collected from each pig for whole genome mRNA sequencing. In addition, blood samples were collected from group 2 pigs on day 7 to replace samples on day 0, as the original samples collected on day 0 were accidentally mishandled and spoilt during the process.

### 2.2. Specific Cellular Immune Response

To define and compare the T cell expansion stimulated by these antigen combinations, CD4+, CD8+, and CD4+/CD8+DP T cell ratios in PBMC samples, collected on day 0 (day 7 for group 2), 28, and 35, respectively, after priming, were measured by flow cytometry using specific antibodies. Results for individual pigs in Group 1 and Group 2 were presented in Figure 2a,b, respectively. In Group 1, each subtype T cell ratio was obviously increased on day 28, compared to that on day 0, but was largely unchanged after the second boost (day 28 compared with day 35), except in the #2 pig which showed much higher CD8+ and CD4+/CD8+DP T counts and much lower CD4+T count on day 28 and 35. It indicates that immunization of the replicon-based DNA (#1 and #2 pigs) or IRP (#5 and #6 pigs) induced prominent T cell expansion shortly after immunization and remained relatively high CD8+ and CD4+/CD8+DP T cell ratios (Figure 2a). In group 2, the total T cell numbers between day 7 and day 28 were not obviously different and a general reduction of T cell counts on day 35 was also observed, except in the #7 pig which showed a very low total T cell count at day 7 and did not reach the peak before day 35 (Figure 2b). Comparing the two groups, the average numbers of total T cells in group 2 were, in general, higher than those in group 1 on both day 28 (67.9% to 65.7%) and 35 (61.3% to 52.3%), and higher CD4+ counts (40.7% to 34.5%) and higher CD4+/CD8+ DP T cell ratios (10.1% to 7.2%) remained at day 35 (Figure 2a,b and Table 1). However, due to large variations of the counts and ratios in individual pigs and the small sample size in each group, the statistical significance of the ubiquitin-antigen fusion on the T cell expansion could not be analyzed.

ELISpot analysis, an enzyme-linked ImmunoSpot assay, was then used to measure the IFN-γ-secreting cells which is important in regulation of the innate and adaptive immunity by antigen stimulation. IFN-γ responses in vitro revealed a correlation with the degree of cross-protection against heterologous ASFV challenge [30]. PBMCs were stimulated by co-incubation with purified p30, p54, and CD2v proteins, respectively, and IFN-γ secreting cells were scored with an ELISpot reader, showing various levels of IFN-γ secreting cells after two boosts. A moderate induction (2-87 IFN-γ secreting cells per 1 million PBMCs) was elicited in #1, #2, #5, #8, and #11 pigs on day 28 with cognate ligands (Figure 2c), and a more diverse level of the induction (2–200 IFN-γ secreting cells per 1 million PBMCs) was detected in all pigs on day 35 (Figure 2d). The fact that IFN-γ secreting cells were detected in #7 and #12 pigs only on day 35 suggests the requirement for a second boost in some recipients (Figure 2c,d).

### 2.3. Specific Humoral Immune Response

The anti-p30 specific antibody levels in serum samples were measured using an ASFV Sandwich ELISA kit, revealing a high level of anti-p30 response in #1 pig just 14 days post-priming (Figure 3a). The antibody level in this pig reached the peak in 2 weeks post-second boost and decreased slowly but remained at a relatively high level throughout the duration of the experiment (Figure 3a). A relatively high antibody induction was detected in the #11 pig from day 14 and steadily increased during the whole observation period, although the level was much lower than that in the #1 pig (Figure 3a). Other antibody-positive pigs included #2, #7, and #8, but the titers were just above the negative control (Figure 3a). It was uncertain if these levels of reaction would provide significant protection against ASFV. Meanwhile, determination of the anti-p54 response by using a competitive ELISA assay showed a 65–96% inhibition in #1 pig from day14 to 70, but other pigs showed less than 50% inhibition (Figure 3b), suggesting that the anti-p54 response was efficiently induced in pig #1 only. Taken together, these results demonstrate that specific humoral response was elicited in 5 out of the 8 immunized pigs, and the earliest date for positive anti-p30 detection was at day 14 post-priming.

### 2.4. Transcriptomic Analysis of DEGs Associated with the Immune Response to the Vaccination

To gain insights into the molecular mechanisms underlying the host immune responses to the vaccination, we analyzed the transcriptomic profiles of whole blood cells from each vaccinated pig. As transient innate responses may reach the peak at 24 h post-immunization and the focus of this study was to characterize the innate immune response from memory cells, we collected the samples immediately before (on day 28) and 24 h post-the second boost (on day 29), respectively, for whole transcriptome sequencing. To confirm the assay reliability, the principal component analysis (PCA) was performed with the data before further biological analysis, showing good technical reproducibility for triplicated samples but poor biological reproducibility due to individual divergence (Appendix A). Differentially expressed genes (DEGs) from day 28 versus day 29 were then identified with statistical verification, revealing a response of genes with great variations in both quantity and characteristics in individual pigs. A total of 274 to 1216 DEGs with different ratios of up- (red) and down-regulated (green) expression was found in different pigs (Figure 4a).

To focus on the immune system, genes not relevant to the immune system were filtered out, and analysis of these immune system-related DEGs (Appendix A) by Venn diagram assay identified 10 common DEGs in all samples (Figure 4b). Interestingly, 9 out of the 10 common DEGs were down-regulated, and one (TGM3) varied in different pigs with the range of FC values indicated in Table 2. These common DEGs include lymphocyte regulator CTLA4, CXC chemokine family members, transcription regulators involved in cell growth, differentiation, or death, and undetermined genes. Furthermore, analysis of the top 20 DEGs from each pair of experimental pigs revealed their diverse distribution, but the five common down-regulated genes with known functions, CXCL2, CXCL8, FOS, CTLA4, and EGR1, were consistently detected (Appendix A). These genes also showed relatively high fold-change and their differential expression appeared not to be affected by the vaccine dosage and form. The genes are therefore the most responsive genes to the boost.

### 2.5. Hub Genes and Molecular Signatures Correlated to the Adaptive Immune Response

Weighted correlation network analysis (WGCNA) has been increasingly used in bioinformatics applications, for finding clusters (modules) of highly correlated genes, for summarizing such clusters using the module eigengene or an intramodular hub gene, for relating modules to one another and to external sample traits (using eigengene network methodology), and for calculating module membership measures [31]. Using this method, we identified the genes associated with the vaccination-stimulated adaptive responses with the transcriptomic data. Taking the anti-p30 antibody OD value measured on day 42 (14 days post-second boost) as a specific trait, we identified a module with an average correlation value of 0.491 and a *p*-value of 0.0148, representing a hub module with a high possibility of positive correlation to the trait (Figure 5a). Genes in this module were then used to identify gene clusters in individual animals through Venn analysis, and further KEGG analysis revealed the most significant gene clusters (*p* adjust value < 0.05) were enriched in the IL17 signaling pathway, C-type lectin receptor (CLR) signaling pathway, NOD-like receptor signaling pathway, Toll-like receptor (TLR) signaling pathway, cytosolic DNA-sensing pathway, T cell receptor signaling pathway, Th17 cell differentiation, RIG-I-like receptor signaling pathway, chemokine signaling pathway, and Th1 and Th2 cell differentiation (Table 3 and Appendix A).

To identify the hub gene and molecular signatures correlated to the cellular immune response, we input the IFN-γ-ELISpot results as the trait and discovered three modules with relatively high correlation coefficients from the total 9 modules. These include a hub module with the highest correlation value [−0.546, *p*-value = 0.00578] corresponding to CD2v-stimulated IFN-γ secretion, a module with a moderately high correlation value (−0.412, *p* value = 0.0454) corresponding to p30-stimulated IFN-γ secretion, and a module with a correlation value (−0.263, *p* value = 0.214) corresponding to p54-stimulation (Figure 5b). All these correlation coefficients in the selected modules were negative, indicating that they are negative regulators correlating to the trait. The first two modules were in the MEturquoise gene-set, the same gene-set correlated to the p30-stimulated antibody production, indicating that genes in these modules may be involved in positively regulating antibody response, but negatively regulating antigen-stimulated IFN-γ secretion. Corresponding to the p54-stimulated IFN-γ secretion, the module in the MEblue gene set showed a slightly higher correlation value (−0.263) than that in MEturquoise (−0.181) (Figure 5b). However, both correlation coefficients were low and the *p*-value were >0.05, raising uncertainty about their biological significance. Although IFN-γ-ELISpot is often used to measure T cell response, it cannot reflect T cell response accurately as NK cells are also a vital source of IFN-γ in PBMCs. The IFN-γ-ELISpot readout implicated a cellular response, involving both T and NK cells, and the use of CD3+ T cells to measure the T cell response may provide a more specific result.

### 2.6. Personalized Analysis to Denote the Genes Involved in Specific Regulation of the Adaptive Immune Response

As the most efficient antibody response was elicited in #1 and #11 pigs, DEGs with *p* adjust < 0.05 in the trait-correlated genes groups (Appendix A) from these two pigs were further analyzed, showing a group of genes, including TLR4, DDX58, DHX58, ZBP1, PTGS2, NFKBIA, TNF, BCL3, IL18, and IL1RAP were up-regulated in #11 pig, but down-regulated in #1 pig (Appendix A, Table 4). Furthermore, no differential transcription of TLR4, DDX58, DHX58, and ZBP1 was observed in #2 and #12 pigs, the two pigs received the same immunization as #1 and #11 pigs, respectively, but did not elicit an efficient antibody response (Appendix A). This expression pattern highlighted that TLR4, DDX58, DHX58, and ZBP1 signaling pathways may play an important role in regulating this vaccine-induced antibody production. Up-regulation of these genes in the pig with a lower antibody level and down-regulation in the pig with a higher antibody level indicates a potential feedback regulatory loop formed around these genes.

Genes important for regulating the cellular response were identified from the trait-correlated genes in #5, #7, and #8 pigs as they elicited more prominent IFN-γ secreting, mainly including a group of up-regulated genes (S100A8, C4BPA, PLAUR, PLAU, A2M, IL18, PIK3CB, CCR1) and a group of down-regulated genes (CXCL2, FOS, IL17B, IFN-ALPHA-15, CTLA4) (Appendix A, Table 4). In addition, as only the #7 pig showed T cell expansion in response to the last boost, unique DEGs in this pig were possibly correlated to this trait. A total of 14 DEGs in the immune system cluster was identified, with about half of them relevant to the CD8+ T cell expansion and/or differentiation by the functional annotation query (marked by “a” superscript in Table 5). All DEGs in this group were up-regulated, except for one uncharacterized gene.

## 3. Discussion

In this study, we show that immunization of a small number of non-inbred pigs with an SFV replicon-based plasmid DNA or replicon particles expressing ASFV antigens elicits efficient cellular and humoral immune responses in some but not all of the immunized pigs. This inconsistency is due to multiple factors, but the complicated genetic background of the non-inbred experimental pigs and the manual delivery method may be the two main ‘culprits’. A much larger-scale animal experiment would be effective to generate statistically significant data and to reduce the variations, but was hampered by many constraints. Alternatively, personalized analysis of immune response to vaccination in individuals would be meaningful to understand the regulatory mechanisms controlling the immune response in animals with limited resources and time. For the same consideration, a ‘vector-only control’ group was not included in this study. Instead, a strategy to compare and analyze data between pre- and post-immunization (post-boost) in individual animals was adopted.

ASFV structure proteins p30, p54, and CD2v are highly immunogenic using different expression systems, such as baculovirus and adenovirus vectors [14,41,42]. Previous studies with an alphavirus replicon system demonstrated that immunization of pigs with P30 replicon particles induced high anti-p30 antibodies after boost with a naturally attenuated ASFV isolates, OURT88/3 [43], but it was unclear if using IRP alone would elicit an efficient adaptive immune response. In this study, we observed quick T cell differentiation and expansion, efficient antibody production, and effective cellular response in several immunized pigs, demonstrating that the SFV-replicon DNA and/or IRP expression system could indeed induce specific humoral and cellular response efficiently without boost with LAV. It points to the possibility of applying the SFV-replicon system as a useful platform for vaccine development against ASFV. The large amount of production of mRNA, ssRNA, dsRNA, and the targeted protein during the one-cycle replication process may act as PAMPs to drive the innate immune response efficiently. These RNA adjuvant properties and higher levels of antigens produced by the self-replication system would make it a more ideal choice over the conventional CMV promoter-based plasmid DNA system.

Five signaling pathways, including TLR, CLR, NOD-like receptor, cytosolic DNA sensor-mediated, and IL17 receptor-mediated signaling pathways, are shown to be positively correlated to the humoral immune response presented by anti-p30 antibody production and inversely correlate to both p30- and CD2v-stimulated cellular response, highlighting their importance in the navigation of the antigen-stimulated adaptive immunity. Both extracellular and intracellular membrane-associated PRRs, including TLR4, DHX58/DDX58 and ZBP1, were mainly identified in pigs with high levels of specific antibody response (#1 and #11 pigs), but barely detected in other pigs after the second boost. TLR4, DHX58/DDX58 and ZBP1 together with their downstream IRF7 and IRF1, the important type1 IFN regulators, may form a dynamic feedback loop. Antigen-stimulated up-regulation of this group of genes would activate the corresponding signaling pathways and induce the expression of cytokines and chemokines required for the activation and proliferation of T cells essential for an efficient humoral response. An overexpression of these cytokines and chemokines, on the other hand, would induce harmful inflammatory response. As a feedback mechanism, the expression of these PRRs and signaling molecules would be repressed, as observed in the #1 pig, to benefit the development of specific cellular immunity as suggested by WGCNA. In support of this, down-regulation of these genes was not observed in #6 and #11 pigs, the two with much lower IFN-γ secreting cells at day 35. As TLRs were able to activate the innate immune system and subsequently lead to effective adaptive immunity, various TLRs have been added as immune modulators in many subunit vaccines to improve their immunogenicity. TLR4 agonists have emerged as highly potent activators of innate immunity in a number of vaccine adjuvants, such as in the approved vaccines Shingrix, Cervarix, and Fendrix [44]. Our results reported here would support that TLR4 and other PPRs may function as key factors in the induction of innate immunity, which subsequently regulates the adaptive immune response induced by the SFV-replicon-based vaccine candidates. Understanding the reciprocal regulation of the two arms of adaptive immunity and the feedback regulatory mechanisms in these pathways would be particularly important for the rational design of vaccines and vaccination programs against ASFV. In addition, ubiquitination of antigens was reported to reduce the efficiency of antibody production and direct the immune system towards cellular immunity with an unknown mechanism [21,23]. In consistency, our study revealed that pigs (#1, #11) elicited higher antibody responses had much lower IFN-γ production, and pigs (#5, #7, #8) elicited lower antibody responses and had higher IFN-γ production. It seems that, however, ubiquitination of the three antigens in this study was not shown to play a significant role in directing the immune response toward cellular immunity as reported.

The complement system is activated by antibodies while activated complement fragments impact the production of antibodies [45]. The complement system is traditionally considered to be an integral part of the innate immune defense against pathogens, playing a main function in lysis of targets such as bacteria. In the context of viral infections, however, the complement system has been shown to exhibit numerous antiviral mechanisms via direct neutralization of both enveloped and non-enveloped viruses, and/or the promotion of other immune responses [45]. Early studies have shown that complement-mediated lysis performed by neutrophils might act as a significant effector and play a role in the recovery from ASFV infection [46,47]. In this study, we observed that CIQA, CIQB and CIQC were significantly up-regulated in response to the boost in most pigs. CIQA, CIQB and CIQC encode the A-, B- and C-chain polypeptides of serum complement subcomponent C1q, respectively, and the hetero-multimer C1q binds to IgG and IgM to initiate the classical complement activation pathway. The accumulation of C1q together with the specific antibody in the immunized pigs may contribute to fast clearance of ASFV and to prevent viral spread.

IL17 signaling pathway modulates the production of cytokines, chemokines, and anti-microbial molecules, and regulates Th-17 cell differentiation, neutrophil recruitment, and T and B cell sensitization. IL17B is potentially involved in the activation of NF-kB and induction of pro-inflammatory cytokine IL8, the gene product of CXCL8 [48], and induces the production of TNFα and IL1β in a monocytic cell line [49]. This study revealed a group of DEGs including IL17B, TNFAIP3, MAPK14, CXCL2, FOS, FOSB, TNF, and CASP3 were significantly down-regulated whereas S100A8, NF-κBIA, and PTGS2 were up-regulated following the second boost. It was reported that inoculation with attenuated genotype I ASFV strains (NH/P68, OURT88/3) induced enhanced expression of key cytokines (IFNβ, several IFNα subtypes, IL1β, IL12p40, TNFα) and chemokines (CCL4, CXCL8, CXCL10) compared with highly virulent isolates (L60, Benin 97/1, 22653/14, Armenia2007) in in vitro cultured macrophage cells [50,51,52,53,54]. These soluble mediators released by the infected macrophages may promote the development of effective adaptive immune responses [18]. However, this study showed that the down-regulation of both CXCL2 and CXCL8 was detected in all vaccinated pigs, with variable levels (either up, down, or no change) of CXCL10 and CCL4. Further studies would be required to clarify the functional involvement of these genes in the host immune response against ASFV.

CLR signaling pathway, including DC-SIGN (CD209) and CLEC family members CLEC4K, CLEC5A, CLEC7A, CLEC9A, CLEC2, CLEC12A, and CLEC12B, is crucial for tailing immune response to pathogens and induces diverse immune responses. DC-SIGN on myeloid DCs is involved in inducing the differentiation of Th1, Th2, Th17, and regulatory T cells through binding to different ligands [55]. CLEC7A expressed in myeloid DCs, monocytes, macrophages, and B cells induces the differentiation of Th1 and Th17 cells through the induction of IL1β, IL6, IL12, and IL23 as well as TNF and CXCL2 [55]. In this study, LEC7A together with the co-factors and transcription products promoted by CLR signaling pathway, such as MAPK14, CASP1, TNF, RELB, EGR2, BCL3, PTGS2, were differentially regulated in several pigs following the second boost. DC-SING(CD209), a DC cell receptor to initiate the adaptive immune response, was highly up-regulated (Log2FC > 3.9) in #1 and #11 pigs, raising a possibility that it may play a role in the regulation of antibody production in these pigs. BCL3 is a main regulator for Th2 cell differentiation, and RELB, MAPK14, CASP1, TNF, PTGS2, and EGR2 are known to be involved in Th1 and Th17 differentiation. Similarly, variable expression levels of these DEGs were found in different pigs, suggesting a dynamic regulatory mechanism at the gene expression level to dictate a different dominance of the Th1 or Th2 response in individual animals.

Down-regulation of CTLA4 and up-regulation of SOCS3 may play a role in this vaccination-stimulated adaptive immunity. CTLA4 is induced in both CD4+ and CD8+ T cells upon T cell activation and functions as a receptor transmitting an inhibitory signal to T cells to act as an immune attenuator [56]. Down-regulation of CTLA4 identified in all vaccinated pigs in this study would prevent the CTLA4-mediated attenuation of T helper cells. SOCS3 is a member of the SOCS (Suppressors of Cytokine Signaling) family. The two types of SOSC family genes (type I, SOCS4-7; type II, CISH, and SOCS1-3) are key negative regulators of cytokine and growth factor signaling and modulators of the T cell biology [57]. In this study, SOCS3 was generally up-regulated and both EGR1 and EGR2 were down-regulated. Further investigation into the potential interaction between these molecules and their biological significance would shed new light on their involvement in regulating the immune response.

In conclusion, we have demonstrated that SFV replicon-based vaccine candidates expressing a combination of ASFV proteins with or without fusion to ubiquitin are highly immunogenic, inducing rapid T cell expansion/activation, specific antibody response, and antigen-induced IFN-γ secretion from PBMCs. However, the efficiencies of such vaccination were heavily influenced by the genetic background and other unknown intrinsic factors of the immunized hosts, and perhaps an inconsistent antigen delivery. Through personalized analysis, a number of trait-relevant hub genes, molecular signaling pathways, and regulatory mechanisms were uncovered. These findings would be instrumental to guiding future systematic studies using a larger scale of animal experiments, inclusion of more antigens contributed to the protective immunity against ASFV (such as EP153R, MGF505-7R and M448R), a combinatorial vaccine regimen capable of inducing both cellular and humoral immune responses systemically, and viral challenge. In addition, optimization of the delivery route and methods would be essential. Considering the richness of antigen-presenting Langerhans cells (LC) and dermal dendritic cells (DDC) in the skin, intradermal (ID) delivery using microneedles may improve the delivery efficiency and consistency.

## 4. Materials and Methods

### 4.1. Antibodies, Chemicals, Reagents, and Cells

Antibodies and reagents were purchased from the following companies: antibody against Flag-tag (#2044, Cell Signaling Technology, Danvers, MA, USA), goat anti-swine IgG (H + L) HRP and IFTC-conjugated AffiniPure goat anti-swine IgG (H + L) (Jackson ImmunoResearch, West Grove, PA, USA), ASFV sera (China Veterinary Drug Administration, Beijing China), FITC-conjugated goat anti-mouse IgG (Li-COR Biosciences, Lincoln, NE, USA), TransIntroTM EL transfection reagent (Transgen, Beijing, China), ClonExpress Multis One Step Cloning Kit (Vazyme Biotech Co., Nanjing, China), anti-Flag gel (#P2282) and 3xFlag (#P9801) (Beyotime, Shanghai, China), CD4 monoclonal antibody (#74-12-4) and FITC-conjugated CD8-α monoclonal antibody (#76-2-11) (ThermoFisher, Waltham, MA, USA), pig lymphocyte isolates (Shenzhen Dakwei Biotechnology Co., LTD, Shenzhen, China), ASFV sandwich ELISA antibody detection kit (Zhaoqing Dahuanong Biological Pharmaceutical Co., LTD, Zhaoqing, China), AsurDxTM ASFV p54 antibody test kits (BioStone, Dallas, TX, USA), porcine IFN-γ ELISpotPLUS kit (HRP) (Dakewe Biotech Co., Ltd., Shenzhen, China), DMEM, RPMI 1640, fetal calf serum (FBS), penicillin and streptomycin (Invitrogen, Waltham, MA, USA), E.Z.N.A. Endo-Free Plasmid Mini Kit (D6948-01) (Solarbio, Beijing, China). pSFVCs-LacZ and pSFV-helper1 (#92076, #92073, Addgene, Watertown, MA, USA).

BHK-21 cells were cultured at 37 °C with 5% CO_2_ in DMEM supplemented with 10% FBS and 1% penicillin-streptomycin. PBMCs were prepared by separation from whole blood through density-gradient centrifugation with pig lymphocyte isolates according to the manufacturer’s protocol. PBMCs were cultured in RPMI 1640 supplemented with 10% FBS, 50,000 IU penicillin/L, and 50 mg streptomycin/L. Trypan blue staining was used to assess the cell viability.

### 4.2. Plasmid Construction

SFV replicon-based ASFV specific antigen expression plasmid SFV-flagP54 was constructed by replacing the SP6 promoter in pSFVCs-lacZ (Addgene) with the human cytomegalovirus immediate-early enhancer and promoter sequences (CMV promoter) and the Cs-lacZ sequence with Flag-tagged ASFV p54 by multi-fragment recombination. The flagP54 sequence was synthesized with BamHI and SmaI sites flanked the p54 sequence. Plasmids SFV-flagCD2v, SFV-flagp30, SFV-flagUbiCD2vEDp30, and SFV-flagUbiCD2vEDp54 were constructed by replacing the p54 gene with synthesized DNA fragments by homologous recombination with BamH1/SmaI-digested pSFV-flagp54. The ubiquitin-fused constructs, UbiCD2vEDp30 and UbiCD2vEDp54, would express human ubiquitin (Ubi)-extracellular domain of CD2v (CD2vED)-p30 and Ubi-CD2vED-p54 chimeric proteins, respectively. All ASFV-specific sequences were designed based on the ASFV_HU_2018 strain sequence (GenBank: MN715134.1). pSFV-helper1-CMV was obtained by modification of pSFV-helper1 (Addgene) with the SP6-promoter sequence replaced by the CMV promoter. All plasmids were verified by sequencing.

### 4.3. Preparation of Single Cycle IRPs

Single-cycle IRPs were prepared by co-transfecting BHK21 cells with pSFV-helper-CMV and an antigen-expressing replicon plasmid, pSFV-flagP54, pSFV-flagCD2v, pSFV-flagp30, pSFV-flagUbiCD2vEDp30 or pSFV-flagUbiCD2vEDp54, using TransIntroTM EL Transfection Reagent according to the manufacturer’s instruction. At 30 h post-transfection, the culture medium was collected for IRP preparation, and cells were lysed for Western blot or protein purification.

After removal of cell debris from the culture medium by brief centrifugation, the IRP titer was determined with a fluorescent antibody staining technique, and the TCID50 (FFU/mL) was calculated using the Reed-Muench method.

### 4.4. SDS-PAGE and Western Blot Analysis

Approximately 25 µg of total proteins from each sample were mixed with LaemmLi’s sample buffer (0.3125 M Tris-HCl, pH 6.8, 10% SDS, 50% glycerol, 25% β-mercaptoethanol, 0.025% bromophenol blue), and separated on an SDS-12% polyacrylamide gel by electrophoresis, followed by transferring onto a nitrocellulose membrane. The membrane was blocked with 5% BSA in TBST (20 mM Tris-HCl pH 7. 4, 150 mM NaCl, 0.1% Tween 20) for 1 h at room temperature, and then incubated with primary antibodies (anti-flag or ASFV serum) at 4 °C overnight. After 3 washes with TBST, the membrane was incubated with 1:15,000 diluted IFCT-conjugated goat anti-mouse IgG or goat anti-swine IgG at room temperature for 1 h. After 3 washes with TBST, the proteins on the membrane were detected with an Azure C600 imager according to the manufacturer’s instructions.

### 4.5. Animal Experimental Design and Immunization

Animals were obtained from the Experimental Animal Center of Xinxing Dahua Agricultural, Poultry and Egg Co., Ltd., Yunfu, China. approved number SCXK (Guangdong) 2018-0019. Animal experiments were performed under the approval of the Animal Welfare and Ethical Censor Committee at South China Agricultural University. After the completion of experimental procedures, all animals were euthanized before mechanistic execution.

Eight 30-day-old Sanhua pigs (hybrid of civil pig Dulock and Big Yorkshire) were divided into two groups and fed for 1 week before immunization. Four pigs in Group 1 were immunized with mixed plasmids or IRPs expressing p54, pCD2v, and p30. Two pigs (#1 and #2) in this group were inoculated with 0.6 mg of total plasmid DNA in saline composed of 0.2 mg each of pSFV-flagP54, pSFV-flagCD2v, and pSFV-flagP30; the other two (#5 and #6) were immunized with 1.8 mL of IRP mixture composed of 0.6 mL (10^6^ FFU/mL) each of rSFV-flagP54, rSFV-flagCD2v, and rSFV-flagP30. The four pigs in group 2 were immunized with mixed plasmids or IRPs expressing UbiCD2vEDp30 and UbiCD2vEDp54. Two pigs (#7 and #8) were immunized with 0.6mg of total plasmid DNA in saline containing 0.3 mg each of pSFV-flagUbiCD2vEDp30 and pSFV-flagUbiCD2vEDp30; pigs #11 and #12 were immunized with 1.2 mL of IRP mixture containing 0.6 mL (10^6^ FFU/mL) each of rSFV-flagUbiCD2vDEp30 and rSFV-flagUbiCD2vEDp54.

The immunization protocol included one priming and two boosts at a 2-week interval, by inoculating one third each of the doses at three different sites (the rectus femoris quadriceps muscle, the trapezius muscle of the neck, and at an ear by a subcutaneous injection). Whole blood samples were collected at 7- to 14-day intervals before inoculation if required.

### 4.6. Measurements of Specific Humoral and Cellular Immune Response

ASFV-specific antibodies in immunized pig sera were measured in duplicate by ELISA assay using the commercially available ASFV sandwich ELISA p30 antibody detection kit and AsurDxTM ASFV p54 Antibodies Test Kit according to the manufacturer’s instructions.

IFN-γ ELISpot kit was used to measure the trained cellular response. Briefly, flagP30, flagP54, and flagCD2v proteins were prepared by anti-Flag gel purification from BHK cells transfected with the respective plasmid using the 3X Flag competitive elution method (Beyotime). Non-denaturing proteins without contamination of light and heavy chains from the Flag antibody was obtained with high elution efficiency. After removal of the 3X Flag peptide by filtration with a protein concentrator (10 K MWCO, Thermo) and washing twice with PBS, the purified proteins were resuspended in PBS. 10^4^ live porcine PBMCs cells from each sample were stimulated for 20 h with 60μg each of the purified flagP30, flagP54 protein, or flagCD2v protein. RPMI1640 only and 10 g/mL of phytohem agglutinin were used as the negative and positive controls, respectively. Samples were sent to BioStone for score and the frequency of the antigen-specific IFN-γ-secreting cells (IFN-γ-SC) per million PBMCs was calculated after subtracting the spot counts in the negative-control wells from triplicates. 

### 4.7. Flow Cytometry

PBMCs were stained with a CD4 monoclonal antibody (74-12-4), FITC and CD8-α monoclonal antibody (76-2-11), and P-phycoerythrin according to the manufacturer’s instruction, and analyzed by flow cytometry (CytoFLEX SRT, Backman Coulter Inc., Indianapolis, IN, USA). The measurement was repeated twice and similar results were obtained, one of them was presented.

### 4.8. Transcriptomic Analysis

Whole blood cells were frozen in TRIzol reagent immediately after collection and sent to Shanghai Majorbio Bio-Pharm Technology Co., Ltd., Shanghai, China for RNA extraction and RNA-seq analysis. Technical triplicates were performed for each vaccinated pig in order to reduce systematic error. Total RNA was extracted from whole blood samples using TRIzol^®^ reagent according to the manufacturer’s instructions (Invitrogen) and genomic DNA was removed using DNase I (TaKara). The RNA quality was determined by 2100 Bioanalyser (Agilent) and quantified using the ND-2000 (NanoDrop Technologies, Wilmington, DE, USA). Only high-quality RNA sample (OD260/280 = 1.8~2.2, OD260/230 ≥ 2.0, RIN ≥ 6.5, 28 S:18 S ≥ 1.0, >1 μg) was used to construct sequencing libraries.

RNA-seq transcriptome libraries were prepared following TruSeqTM RNA sample preparation with the Illumina Kit (San Diego, CA, USA) using 1μg of total RNA. Total mRNA was isolated with oligo (dT) beads and fragmented with a fragmentation buffer, and double-stranded cDNA was synthesized using a SuperScript double-stranded cDNA synthesis kit (Invitrogen, CA, USA) with random hexamer primers (Illumina). The synthesized cDNA was then subjected to end-repair, phosphorylation and ‘A’ base addition. Libraries were size-selected for cDNA target fragments of 300 bp on 2% low-range ultra-agarose separation gel after PCR amplification for 15 cycles using Phusion DNA polymerase (NEB). After quantified by TBS380, the end-paired RNA-seq libraries were sequenced with an Illumina HiSeq xten/NovaSeq 6000 sequencer (2 × 150 bp read length).

The raw paired end reads were trimmed and quality-controlled by SeqPrep (https://github.com/jstjohn/SeqPrep) and Sickle (https://github.com/najoshi/sickle) with default parameters. Clean reads were then separately aligned to reference genomes with orientation mode using HISAT2 (http://ccb.jhu.edu/software/hisat2/index.shtml) software. The mapped reads of each sample were assembled by StringTie (https://ccb.jhu.edu/software/stringtie/index.shtml? t = example) in a reference-based approach.

To identify differentially expressed genes (DEGs) between two different samples, the expression level of each transcript was calculated according to the transcripts per million reads (TPM) method, and RSEM (http://deweylab.biostat.wisc.edu/rsem/) was used to quantify the gene abundance. Essentially, differential expression analysis was performed using the DESeq2 [4]/DEGseq [5]/EdgeR [6] with Q value ≤ 0.05, DEGs with |log2FC| > 1 and Q value ≤ 0.05 (DESeq2 or EdgeR)/Q value ≤ 0.001 (DEGseq) were considered to be significantly different for the expressed genes. The RNA sequence raw data were submitted to NCBI and the access number is PRJNA936467. In addition, functional-enrichment analyses including GO and KEGG were performed to identify which DEGs were significantly enriched in GO terms and metabolic pathways at Bonferroni-corrected *p*-value ≤ 0.05, compared with the whole-transcriptome background. GO functional enrichment and KEGG pathway analysis were carried out by Goatools (https://github.com/tanghaibao/Goatools) and KOBAS (http://kobas.cbi.pku.edu.cn/home.do) to summarize the functional annotation of genes/transcripts in NR, Swiss-Prot, Pfam, EggNOG, GO and KEGG databases, and provide two retrieval methods to query the gene information corresponding to specific functions, or retrieve their functions according to the gene information, so as to quickly lock the required information. More technical details in RNA extraction, library preparation, Illumina Hiseq xten/Nova seq 6000 Sequencing, read mapping, differential expression analysis, principal component analysis (PCA), and functional enrichment are available at www.majorbio.com.

To select genes associated with vaccination-induced adaptive immunity, the weighted correlation network analysis (WGCNA) R software package, a comprehensive collection of R functions for performing various aspects of weighted correlation network analysis, was used [32]. In principle, co-expression networks are undirected weighted gene networks. The node of such a network corresponds to gene expression profiles, and edges between genes are determined by the pairwise correlations between gene expressions. The soft threshold was used to ensure a scale-free network, and Pearson’s coefficient correlations were calculated to assess the relationships between gene expression data and the trait [32]. A software package available at www.majorbio.com was applied for WGCAN in this study. The transcriptomic datasets collected on days 28 and 29 post-priming were auto-filtered through cluster analysis to remove the outlier, and genes/transcripts were divided into modules according to the expression trend of genes/transcripts with this software package. The correlation of modules with specific phenotypes was obtained by inputting the trait (OD values of anti-p30 or IFN-γ ELISpot readouts) in WGCNA. The modules with the highest correlation and more significant *p*-value were selected as the characteristic modules for the phenotype. The hub genes were selected from the characteristic modules by Venn with DEGs in pigs eliciting the most obvious trait. The molecular signatures of the hub genes were denoted with KEGG assay and signaling pathways with *p*-value < 0.05 were considered a significantly enriched term.

## Figures and Tables

**Figure 1 ijms-24-09316-f001:**
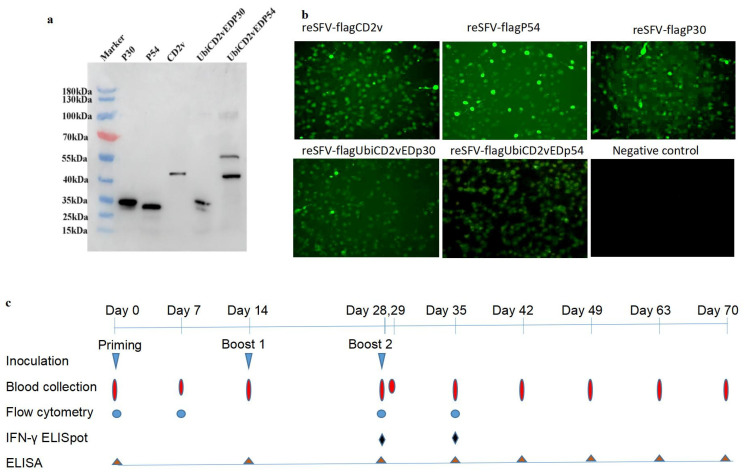
Antigen expression, immunization design and sample collection schedules. (**a**). Expression of the Flag-targeted ASFV structure proteins p30, p54, CD2v, the N-terminal ubiquitin-fused CD2vEDp30 and CD2vEDp54 proteins expressed in BHK cells was detected by Western blot with an anti-ASFV serum. (**b**). Infection of BHK cells with recombinant replicon particles reSFV-flagCD2v, reSFV-flagP54, reSFV-flagP30, reSFV-flagubiCD2vEDP30 and reSFV-flagubiCD2vEDP54, respectively, and the expression of flagCD2v, P54, P30, flagubiCD2vEDP30 and flagubiCD2vEDP54 was detected by immunostaining with and anti-ASFV serum and photographed with fluorescent microscopy. (**c**). A schematic diagram illustrating the schedules of antigen inoculationand sample collection.

**Figure 2 ijms-24-09316-f002:**
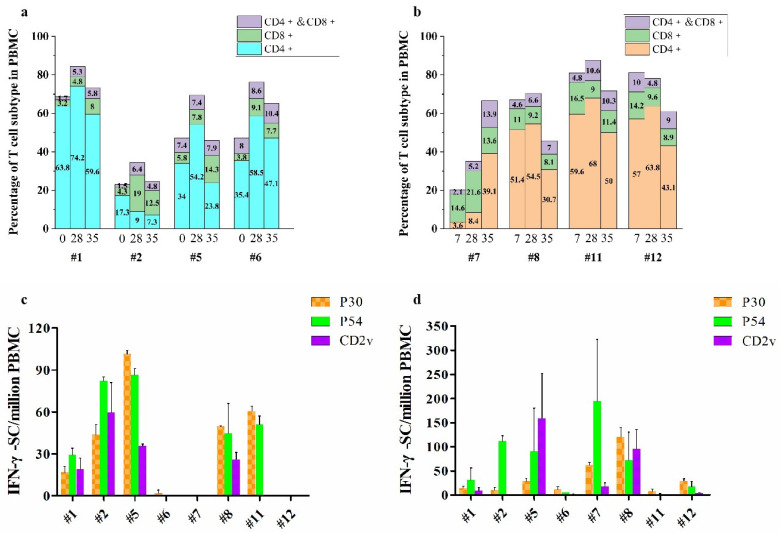
T cell expansion and cellular immune response stimulated by vaccination. (**a**). PBMC cells collected on day 0, 25 and 35 post-priming from group 1 pigs were labeled with relevant surface antibodies and sorted by flow cytometry. The percentages of CD4+, CD8+ and CD4+/CD8+ DPT cells, respectively, in samples were shown. (**b**). PBMC cells collected on day 7, 25 and 35 post-priming from group 2 pigs were analyzed and presented as in (**a**). (**c**). IFNγ-secretory spots (IFN-γ-SC) (mean ± SE) from PBMCs collected on day 28 post-priming and stimulated by p30 (orange), p54 (green) and CD2v (purple), respectively, were determined by ELISpot assay. IFNγ-secretory cell counts were obtained after deducting the basal count (N) from the coordinating sample without stimulation, and control data were obtained by incubation with phytohem agglutinin (yellow). (**d**). IFNγ-secretory spots (mean ± SE) from PBMCs collected on day 35 post-priming and stimulated by p30 (orange), p54 (green) and CD2v (purple), respectively, were determined and presented as in (**c**).

**Figure 3 ijms-24-09316-f003:**
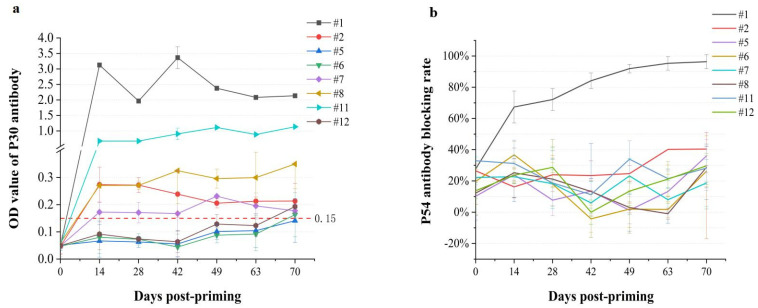
Specific humoral immune response stimulated by immunization. (**a**). Anti-p30 antibody levels (mean ± SE) in sera collected from immunized pigs on day 0, 14, 28, 42, 49, 63 and 70, respectively, were detected with ELISA kit. An OD value of 0.15 or above is considered positive according to the manufacturer’s instruction. (**b**). Anti-p54 antibody (mean ± SE) levels in sera collected from immunized pigs on day 0, 14, 28, 42, 49, 63 and 70, respectively, were determined with a competitive ELISA kit. An inhibition rate of 50% or above is considered positive according to the manufacturer’s instruction.

**Figure 4 ijms-24-09316-f004:**
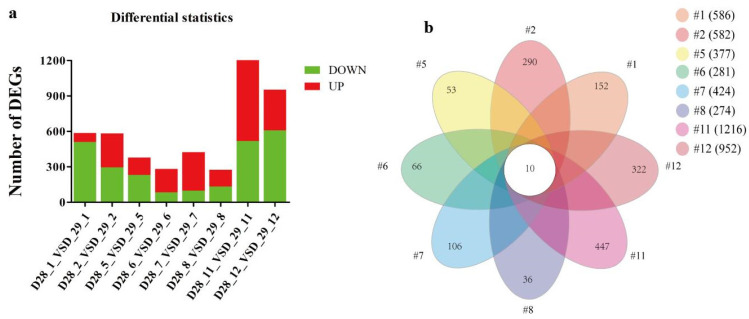
DEGs distribution in vaccinated pigs. (**a**). Profiles of deferentially expressed genes (DEGs) on day 28 versus day 29 from each vaccinated pig were presented. Counts for up-regulated DEGs are marked in red and down-regulated DEGs in green. (**b**). DEG counts for each individual pigs and overly shared DEGs in the immune system were analyzed by Venn and presented.

**Figure 5 ijms-24-09316-f005:**
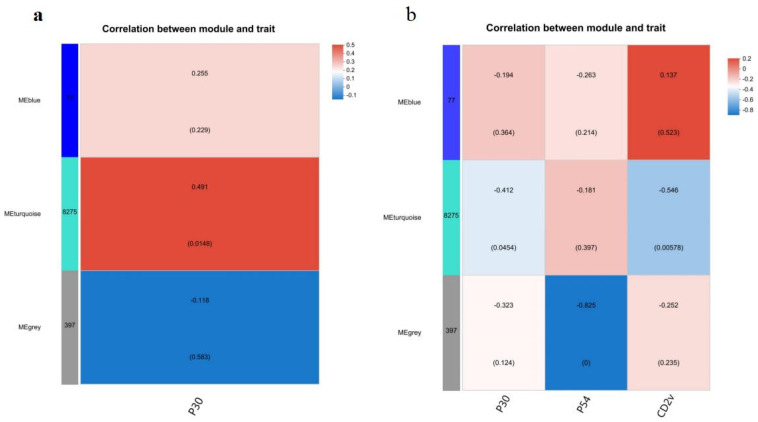
Correlation between DEGs and the trait by WGCNA. (**a**). Correlation between gene clusters and the anti-p30 induction trait. The red module classified in the MEturquoise gene cluster represents average correlation value (0.491) and *p* value (0.0148), the pink module classified in the MEblue gene cluster represents correlation value (0.255) and *p* value (0.229), and the MEgrey gene cluster contains non-correlated genes. (**b**). Correlation between gene clusters and the ELISpot count trait. Modules classified in the MEturquoise gene cluster include the light blue module with the correlation value −0.546 and *p* value 0.00578 to genes responding to CD2v-stimulated IFNγ-SC counts, the green module with the correlation value −0.412 and *p* value 0.0454 to genes responding to p30-stimulated IFNγ-SC counts and the pink one with the average correlation value −0.181 and *p* value 0.397 to genes responding to p54-IFNγ-SC counts. Modules classified in the MEblue gene cluster represents a relative low correlation value and large *p* value (>0.05), and MEgrey contains non-correlated genes. The total DEGs in individual pigs are shown in bracket.

**Table 1 ijms-24-09316-t001:** Average percentage of CD4+, CD8+, CD4+/CD8+ DP and total T cells in PBMC samples.

T Cell Subtype	Group 1			Group 2		
	Day 0	Day 28	Day 35	Day 7	Day 28	Day 35
CD4	37.6%	49.1%	34.5%	42.9%	49.3%	40.7%
CD8	4.3%	9.7%	10.6%	14.1%	12.3%	10.5%
CD4/CD8	4.7%	6.9%	7.2%	5.4%	6.3%	10.1%
Total	46.6%	65.7%	52.3%	62.4%	67.9%	61.3%

**Table 2 ijms-24-09316-t002:** Gene description and expression of 10 commonly shared DEGs.

Gene ID *	Gene Name	Description	Expression Range (Log_2_FC)	Function
08959	CXCL2	chemokine (C-X-C motif) ligand 2	−(1.4 to 7.9)	inflammation
08953	CXCL8	C-X-C motif chemokine ligand 8	−(2.2 to 6.2)	inflammation
02383	FOS	Fos proto-oncogene, AP-1 transcription factor subunit	−(1.4 to 6.1)	cell proliferation, differentiation, and transformation
32416	RGS1	regulator of G protein signaling 1	−(2.1 to 5.8)	leukocyte trafficking, vascular inflammation
14336	EGR1	early growth response 1	−(1.2 to 4.4)	transcription factor involve in cell proliferation, differentiation, invasion, and apoptosis
32613	SNAI1	snail family transcriptional repressor 1	−(1.5 to 3.8)	transcription regulator, T cell differentiation
16122	CTLA4	cytotoxic T-lymphocyte associated protein 4	−(1.2 to 2.8)	regulation of T and B cell for adaptive immune response
26043	TGM3	transglutaminase 3	−2.6 to +4.4	response to poly I:C stimulated innate immunity in pig
45263	/	NA	−(1.5 to 3)	NA
48678	/	NA	−(1.1 to 2.6)	NA

* Gene ID: ENSSSCG000000xxxxx.

**Table 3 ijms-24-09316-t003:** Summary of KEGG enrichment assay.

Pathway	Sum of Gene Number								−LOG10 (Padjust)							
	#1	#2	#5	#6	#7	#8	#11	#12	#1	#2	#5	#6	#7	#8	#11	#12
IL-17 signaling pathway	10	7	5	5	9	7	11	11	5.23	3.51	1.49	3.21	4.49	4.06	3.07	3.68
C-type lectin receptor signaling pathway	9	4	1	1	8	3	11	3	4.09	1.23	0.17	0.36	3.22	1.15	2.53	0.19
NOD-like receptor signaling pathway	9	4	4	2	9	7	19	11	3.09	0.85	0.63	0.58	2.87	3.11	5.10	1.91
Toll-like receptor signaling pathway	7	5	3	2	10	6	13	13	2.89	1.54	0.58	0.74	4.86	2.97	3.76	4.47
Cytosolic DNA-sensing pathway	5	3	4	/	6	3	9	4	2.23	1.09	1.34	/	2.77	1.47	2.82	0.54
T cell receptor signaling pathway	6	2	3	3	6	4	7	6	1.62	0.30	0.49	1.10	1.50	1.41	0.48	0.52
Th17 cell differentiation	6	4	2	1	5	2	11	3	1.53	0.88	0.29	0.35	0.96	0.45	1.46	0.11
RIG-I-like receptor signaling pathway	4	/	3	1	4	3	8	6	1.39	/	0.68	0.40	1.29	1.39	1.92	1.23
Chemokine signaling pathway	6	6	4	1	7	4	10	8	1.37	1.42	0.60	0.35	1.70	1.32	0.88	0.81
Th1 and Th2 cell differentiation	4	3	2	1	3	2	8	2	0.77	0.62	0.32	0.36	0.44	0.51	0.80	0.08
Fc gamma R-mediated phagocytosis	3	3	2	1	/	1	3	1	0.64	0.83	0.45	0.39	/	0.31	0.08	0.04
Toll and Imd signaling pathway	2	/	/	/	1	1	1	/	0.63	/	/	/	0.25	0.45	0.02	/
B cell receptor signaling pathway	3	1	2	1	6	3	3	2	0.62	0.21	0.44	0.38	2.18	1.29	0.08	0.11
Hematopoietic cell lineage	3	1	1	2	3	/	9	5	0.52	0.18	0.17	0.73	0.52	/	1.45	0.48
Fc epsilon RI signaling pathway	2	1	1	1	1	1	3	2	0.37	0.24	0.23	0.40	0.17	0.36	0.16	0.16
Antigen processing and presentation	2	1	/	1	1	1	3	3	0.24	0.18	/	0.36	0.10	0.29	0.05	0.19
Intestinal immune network for IgA production	1	/	/	/	2	/	7	2	0.11	/	/	/	0.29	/	0.80	0.10
Natural killer cell mediated cytotoxicity	1	5	3	1	4	5	12	7	0.10	1.50	0.55	0.36	0.88	2.33	2.81	1.02
Complement and coagulation cascades	/	2	5	3	6	3	5	5	/	0.52	1.41	1.61	2.51	1.39	0.60	0.81
Leukocyte transendothelial migration	/	2	1	1	/	1	3	4	/	0.41	0.17	0.37	/	0.30	0.06	0.35
Platelet activation	/	/	1	/	1	4	4	4	/	/	0.15	/	0.09	1.50	0.08	0.23
P32 OD value (Positive: OD ≥ 0.15)	3.37	0.24	0.06	0.04	0.17	0.32	0.90	0.06	3.37	0.24	0.06	0.04	0.17	0.32	0.90	0.06

**Table 4 ijms-24-09316-t004:** The trait-correlated DEGs with Padjust < 0.05 in signaling pathways.

	Up-Regulated	Down-Regulated
#1	CCR5; CX3CR1	TNFAIP3; IL17B; CXCL2; PTGS2; NFKBIA; FOS; FOSB; TNF; BCL3; EGR2; IL18; TLR4; CCL3L1; ZBP1; DDX58; CTLA4; IL1RAP; DHX58; XCR1
#5	S100A8; C4BPA; THBD; PLAUR; PLAU; A2M; IL18; GNG10; PIK3CB; CCR1; IFNGR1; MARCKSL1	CXCL2; FOS; IL17B; ZBP1; DDX58; IFN-ALPHA-15; DHX58; CTLA4
#7	CD14; S100A8; CLEC7A; RELB; IL18; PLAUR; C5AR1; SERPING1; C4BPA; PLAU; A2M; NFKBIE; FGR; CCR1; GNG10; TEC	CCL3L1; IFN-ALPHA-15; CCL4; IFN-DELTA-1; FOS; TNF; CXCL2; FOSB; EGR2; CTLA4
#8	S100A8; PIK3CB; 5_8S_rRNA; PLAUR; PLAU; A2M; CCL14; CCR2	TNFAIP3; CXCL2; FOS; FOSB; IL17B; IFN-ALPHA-15; CTLA4
#11	MAPK14; RIPK3; RNASEL; NLRP12; GP91-PHOX; CASP1; TNF; IL18; TLR4; NOD1; CD14; TLR1; TLR2; TLR6; TLR8; S100A8; PTGS2; CASP3; TREX1; ZBP1; DDX58; IFNGR1; SH3BP2; IFNGR2; GHSR; BCL3; CLEC7A; RELB; DHX58; IFIH1; NFKBIE; IL4R; IL1RAP; IL6R; JAK2; IL1A; CSF1	TNFAIP3; IFN-ALPHA-15; CXCL2; IFN-DELTA-1; FOS; FOSB; IL17B; EGR2; RORA, CTLA4

**Table 5 ijms-24-09316-t005:** Unique DEGs of No.7 pig.

Gene ID *	Gene Name	Function	Ref.
40811	PYCARD	promotes secretion of pro-inflammatory cytokines	[32]
36521	TNFSF13	This protein and its receptor are both found to be important for B cell development. Gene ID: 106736	
13418	CFD	essential for activation of the complement system;participates for complement activation and inflammation. Gene ID: 1675	
07722 ^a^	NCF1	It is subunit of NADPH oxidase (NOX), which plays an essential role in the immune system. NCF1 in dendritic cells promotes autoreactive CD8 + T cell activation	[33]
03578	FGR	FGR is a myeloid Src-family kinase and participates in regulation of acute myeloid leukaemia.	[34]
36317 ^a^	VASP	contributes to naïve CD8 + T cell activation and expansion by promoting T cell-APC interactions in vivo.	[35]
02919	TYROBP	a key regulator in immune systems by acting as a signaling adaptor in dendritic cells, osteoclasts, macrophages, and microglia.	[36]
29521	CD82	a member of transmembrane 4 superfamily and an activation Ag of T-cells. CD82 associates with CD4 or CD8 and delivers costimulatory signals for the TCR/CD3 pathway.	[37]
17920 ^a^	CXCL16	CXCL16 is produced by dendritic cells in T cell of lymphoid organs and induced by IFN-ɤ and TNF-a.This transmembrane chemokine is implicated in activated CD8+ T cell trafficking.	[38]
35226 ^a^	NA	is collected in gene set for CD8 T cell (https://report.majorbio.com/)	
37426 ^a^	NA	is highly enriched genes in myeloid, NK, CD21pB, SWC6gdT, and CD4CD8T-cells.	[39]
05638	LCN2	lipocalin 2, works in antibacterial, anti-inflammatory, and protection against cell and tissue stress.	[40]
36017 ^a^	NA	is collected in gene set for CD8 T cell (https://report.majorbio.com/)	
44758	NA	Undetermined gene	

* Gene ID: ENSSSCG000000xxxxx; “^a^”: genes relevant to CD8+ T cell function by the functional annotation query.

## Data Availability

Data supporting the findings of this study are available in this paper, Appendix A, or are available from the corresponding authors upon request.

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
