# Peer review of "Molecular Signatures in Swine Innate and Adaptive Immune Responses to African Swine Fever Virus Antigens p30/p54/CD2v Expressed Using a Highly Efficient Semliki Forest Virus Replicon System"

_ijms, 2023, doi:10.3390/ijms24119316_

Round 1
Reviewer 1 Report
In animal experimental design the control group is missing; it must be considered a control group.
Author Response
Reviewer 1
In animal experimental design the control group is missing; it must be considered a control group.
Response: A ‘vector-only control’ group was indeed considered during the experimental design, but was not included after due consideration. As a very small number of non-inbred animals was used in each group, it would be very difficult to compare between groups and to generate statistically significant data. Instead, a strategy to compare and analyze data between pre- and post-immunization (post-boost) in individual animals was adopted. As reiterated in the manuscript, this is an inherent limitation of such small-scale animal studies. We have added two sentences to the discussion part to indicate this fact (“For the same consideration, a ‘vector-only control’ group was not included in this study. Instead, a strategy to compare and analyze data between pre- and post-immunization (post-boost) in individual animals was adopted.” (lines 306-308)
Reviewer 2 Report
The authors of this manuscript demonstrate the molecular signatures in the pig immune response to an ASF vaccine designed using a SFV replicon system. Overall, I found the science of the manuscript to be novel and sound, however the grammar needs improvement throughout.
The comments I have regarding this manuscript are as follows:
Line 50: “LAV protected ASF effectively, but is generally…”
· Protected against ASF
Line 52-55: “Inefficient immunization with LAV in some animals could lead to severe dire consequence with fast viral spread, as such vaccination might be misleading and compromise the enforcement of other biosafety 54 measures.”
· Wording is awkward here. What are the “severe dire consequences”?
· Expand/Clarify on what is meant by “vaccination might be misleading.”
Line 55-59: Consider re-wording this long, run-on sentence.
Line 64: summarized what is meant by “controversial observations were also reported.”
Line 67: Change to “DNA vaccines have…”
Line 69-70: Change to “DNA vaccines have….”
Line 71: Change improvement to improvements
Line 90-91: change to outer envelope.
Line 93: “Anti-CDv2 possesses neutralizing activity.”
· Should include the word “antibody” for clarity.
Line 107: “…the SFV replicon-based DNA vaccine candidates…”
· This need to be expanded. There is no information in the introduction about SFV prior to this mention. SFV should be spelled out initially here in the introduction. Please include more information and references on Semliki Forest Virus as a vaccine platform
Line 157: “cunt” should be count.
Line 183-184: “relatively high level within the whole 70 days of the experimental duration”
· Awkward sentence. Consider rewording to “…high level throughout the duration of the experiment.”
Line 187: “but the titers were just above the negative control.”
· Since these levels were just above the negative control, I think it would be important to discuss the biological significance here (eg protective?)
Line 198: Remove the word “mainly.”
Line 201: Change “whole genomic RNA sequencing” to whole transcriptome sequencing since you are looking at RNA.
Line 205: You use the abbreviation DEG here. This should be spelled out the first time not abbreviated. Consider expanding.
Line 206: “robust response of genes”
· Wording needs to be change here. Consider response of gene expression
Line 196 – 208: Consider including details on RNA sequencing such as number of raw reads acquired/animal or number of mapped reads/animal. This would be useful information here.
Figures:
It seems the figure legends were imported as a picture. This should be converted to text, so that the resolution does not look so grainy.
Consider improving the resolution of the graphs to ease interpretation.
Other
· Was challenge with ASF an option here? It would be very informative to see if the vaccine protected against challenge. This would allow direct comparison between immune data and protection status, which would provide extremely useful data.
· It is noted that the efficiency of the vaccine varied greatly among animal in the same group. You mention it could be due to “an inconsistent antigen delivery”. How would you address this in the future? How will address the varied vaccine efficiency in the future, as this is a significant issue in vaccine development?
Moderate editing of English is required though out the manuscript. I left some detailed comments and suggestions, but this is not all inclusive.
Author Response
Reviewer 2
The authors of this manuscript demonstrate the molecular signatures in the pig immune response to an ASF vaccine designed using a SFV replicon system. Overall, I found the science of the manuscript to be novel and sound, however the grammar needs improvement throughout.
Response: We thank this reviewer for carefully reading the manuscript and the constructive comments. In addition to addressing the issues raised by the reviewer, we have also carefully checked and re-edited the grammatical errors throughout.
The comments I have regarding this manuscript are as follows:
Line 50: “LAV protected ASF effectively, but is generally…” to “…protected against ASF.
Response: Changed (Line 51).
Line 52-55: “Inefficient immunization with LAV in some animals could lead to severe dire consequence with fast viral spread, as such vaccination might be misleading and compromise the enforcement of other biosafety measures.” Wording is awkward here. What are the “severe dire consequences”? Expand/Clarify on what is meant by “vaccination might be misleading.”
Response: We have modified the sentence to “Vaccination of pigs with inefficient LAV might compromise the enforcement of other biosafety measures, leading to a fast viral spread.” (lines 53-58)
Line 55-59: Consider re-wording this long, run-on sentence.
Response: We have modified the sentence to “Development of chronic ASF post-LAV vaccination was also reported in pigs vaccinated with a Portuguese ASFV attenuated by serial passaging in bone marrow cell cultures, a naturally attenuated ASFV isolate, ASFV/NH/P68, and a vaccine candidate, OUR T88/3, respectively [7-10].” (lines 62-64)
Line 64: summarized what is meant by “controversial observations were also reported.”
Response: We have modified the sentence to “Neutralizing antibodies against ASFV structural proteins may partially prevent animals from infection and delay disease development, but are unable to completely block viral replication and spread [15-19].” (lines 61-63)
Line 67: Change to “DNA vaccines have…”
Response: We have changed “DNA vaccine has…” to “DNA vaccines have…” (line 67).
Line 69-70: Change to “DNA vaccines have….”
Response: We have changed “DNA vaccine has…” to “DNA vaccines have…” (line 70).
Line 71: Change improvement to improvements
Response: Changed (line 71).
Line 90-91: change to outer envelope.
Response: Changed (line 88).
Line 93: “Anti-CDv2 possesses neutralizing activity.” Should include the word “antibody” for clarity
Response: Changed (line 93).
Line 107: “…the SFV replicon-based DNA vaccine candidates…” This need to be expanded. There is no information in the introduction about SFV prior to this mention. SFV should be spelled out initially here in the introduction. Please include more information and references on Semliki Forest Virus as a vaccine platform
Response: We have added the following sentences to the discussion: “Semliki Forest virus (SFV)-based replicon is derived from SFV, a member in the Alphavirus genus with a positive-stranded RNA genome of approximately 12 kb. The genome contains two main open reading frames (ORF), encoding a viral replicase and structural proteins, respectively. Replacement of the ORF coding for structural proteins by foreign genes creates an SFV-based vector system with a broad host range and high expression efficiency through self-driven replication. As this replication is self-limited due to the lack of viral structure proteins during the replicon package, it is an efficient and safe platform for foreign gene expression and vaccine development.” (lines 105-111)
Line 157: “cunt” should be count.
Response: Corrected (line 162).
Line 183-184: “relatively high level within the whole 70 days of the experimental duration”. Awkward sentence. Consider rewording to “…high level throughout the duration of the experiment.”
Response: Changed (lines 188-189).
Line 187: “but the titers were just above the negative control.” Since these levels were just above the negative control, I think it would be important to discuss the biological significance here (eg protective?)
Response: We have changed the sentences to “…but the titers were just above the negative control (Fig. 3a). It was uncertain if these levels of reaction would provide significant protection against ASFV.” (lines 193-194)
Line 198: Remove the word “mainly.”
Response: Removed (line 204).
Line 201: Change “whole genomic RNA sequencing” to “whole transcriptome sequencing since you are looking at RNA”.
Response: Changed (line 207).
Line 205: You use the abbreviation DEG here. This should be spelled out the first time not abbreviated. Consider expanding.
Response: Modified (line 211).
Line 206: “robust response of genes”. Wording needs to be change here. Consider response of gene expression
Response: Removed “robust” (line 212).
Line 196 – 208: Consider including details on RNA sequencing such as number of raw reads acquired/animal or number of mapped reads/animal. This would be useful information here.
Response: We have submitted the RNA sequence data to NCBI. The access number is PRJNA936467 and added this information in the Material and method (279-280).
Figures: It seems the figure legends were imported as a picture. This should be converted to text, so that the resolution does not look so grainy. Consider improving the resolution of the graphs to ease interpretation.
Response: Original figures and legends will be resubmitted separately, according to the guidelines of the Journal.
Other
- Was challenge with ASF an option here? It would be very informative to see if the vaccine protected against challenge. This would allow direct comparison between immune data and protection status, which would provide extremely useful data.
Response: We agree that challenge with ASFV would provide useful information here, but, unfortunately, we are unable to do it at the moment due to the biosafety and several other constraints.
- It is noted that the efficiency of the vaccine varied greatly among animal in the same group. You mention it could be due to “an inconsistent antigen delivery”. How would you address this in the future? How will address the varied vaccine efficiency in the future, as this is a significant issue in vaccine development?
Respond: We have added the following sentences to the discussion: “In addition, optimization of the delivery route and methods would be essential. Considering the richness of antigen presenting Langerhans cells (LC) and dermal dendritic cell (DDC) in skin, intradermal (ID) delivery using microneedles may improve the delivery efficiency and consistency.” (lines 431-435)
Round 2
Reviewer 1 Report
I have no Comments and Suggestions for Authors.
A minor revision of English language is recommended.